# Generating Long-term Trajectories Using Deep Hierarchical Networks

**Stephan Zheng**
Caltech
stzheng@caltech.edu

**Yisong Yue**
Caltech
yyue@caltech.edu

**Patrick Lucey**
STATS
plucey@stats.com

## Abstract

We study the problem of modeling spatiotemporal trajectories over long time horizons using expert demonstrations. For instance, in sports, agents often choose action sequences with long-term goals in mind, such as achieving a certain strategic position. Conventional policy learning approaches, such as those based on Markov decision processes, generally fail at learning cohesive long-term behavior in such high-dimensional state spaces, and are only effective when fairly myopic decision-making yields the desired behavior. The key difficulty is that conventional models are "single-scale" and only learn a single state-action policy. We instead propose a hierarchical policy class that automatically reasons about both long-term and short-term goals, which we instantiate as a hierarchical neural network. We showcase our approach in a case study on learning to imitate demonstrated basketball trajectories, and show that it generates significantly more realistic trajectories compared to non-hierarchical baselines as judged by professional sports analysts.

## 1 Introduction

Modeling long-term behavior is a key challenge in many learning problems that require complex decision-making. Consider a sports player determining a movement trajectory to achieve a certain strategic position. The space of such trajectories is prohibitively large, and precludes conventional approaches, such as those based on simple Markovian dynamics.

Many decision problems can be naturally modeled as requiring high-level, long-term *macro-goals*, which span time horizons much longer than the timescale of low-level *micro-actions* (cf. He et al. [8], Hausknecht and Stone [7]). A natural example for such macro-micro behavior occurs in spatiotemporal games, such as basketball where players execute complex trajectories. The micro-actions of each agent are to move around the court and, if they have the ball, dribble, pass or shoot the ball. These micro-actions operate at the centisecond scale, whereas their macro-goals, such as "maneuver behind these 2 defenders towards the basket", span

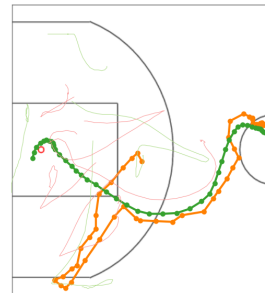

Figure 1: The player (green) has two macro-goals: 1) pass the ball (orange) and 2) move to the basket.

multiple seconds. Figure 1 depicts an example from a professional basketball game, where the player must make a sequence of movements (micro-actions) in order to reach a specific location on the basketball court (macro-goal).

Intuitively, agents need to trade-off between short-term and long-term behavior: often sequences of individually reasonable micro-actions do not form a cohesive trajectory towards a macro-goal. For instance, in Figure 1 the player (green) takes a highly non-linear trajectory towards his macro-goal of positioning near the basket. As such, conventional approaches are not well suited for these settings, as they generally use a single (low-level) state-action policy, which is only successful when myopic or short-term decision-making leads to the desired behavior.

In this paper, we propose a novel class of *hierarchical policy models*, which we instantiate using recurrent neural networks, that can simultaneously reason about both macro-goals and micro-actions. Our model utilizes an attention mechanism through which the macro-policy guides the micro-policy. Our model is further distinguished from previous work on hierarchical policies by dynamically predicting macro-goals instead of following fixed goals, which gives additional flexibility to our model class that can be fitted to data (rather than having the macro-goals be specifically hand-crafted).

We showcase our approach in a case study on learning to imitate demonstrated behavior in professional basketball. Our primary result is that our approach generates significantly more realistic player trajectories compared to non-hierarchical baselines, as judged by professional sports analysts. We also provide a comprehensive qualitative and quantitive analysis, e.g., showing that incorporating macro-goals can actually improve 1-step micro-action prediction accuracy.

## 2 Related Work

The reinforcement learning community has largely focused on non-hierarchical policies such as those based on Markovian or linear dynamics (cf. Ziebart et al. [17], Mnih et al. [11], Hausknecht and Stone [7]). By and large, such policy classes are shown to be effective only when the optimal action can be found via short-term planning. Previous research has instead focused on issues such as how to perform effective exploration, plan over parameterized action spaces, or deal with non-convexity issues from using deep neural networks. In contrast, we focus on developing hierarchical policies that can effectively generate realistic long-term plans in complex settings such as basketball gameplay.

The use of hierarchical models to decompose macro-goals from micro-actions is relatively common in the planning community (cf. Sutton et al. [14], He et al. [8], Bai et al. [1]). For instance, the winning team in 2015 RoboCup Simulation Challenge (Bai et al. [1]) used a manually constructed hierarchical policy to solve MDPs with a set of fixed sub-tasks, while Konidaris et al. [10] segmented demonstrations to construct a hierarchy of static macro-goals. In contrast, we study how one can *learn* a hierarchical policy from a large amount of expert demonstrations that can adapt its policy in non-Markovian environments with dynamic macro-goals.

Our approach shares affinity with behavioral cloning. One difference with previous work is that we do not learn a reward function that induces such behavior (cf. Muelling et al. [12]). Another related line of research aims to develop efficient policies for factored MDPs (Guestrin et al. [6]), e.g. by learning value functions over factorized state spaces for multi-agent systems. It may be possible that such approaches are also applicable for learning our hierarchical policy.

Attention models for deep networks have mainly been applied to natural language processing, image recognition and combinations thereof (Xu et al. [15]). In contrast to previous work which focuses on attention models of the input, our attention model is applied to the *output* by integrating control from both the macro-policy and the micro-policy.

Recent work on generative models for sequential data (Chung et al. [4]), such as handwriting generation, have combined latent variables with an RNN's hidden state to capture temporal variability in the input. In our work, we instead aim to learn semantically meaningful latent variables that are external to the RNN and reason about long-term behavior and goals.

Our model shares conceptual similarities to the Dual Process framework (Evans and Stanovich [5]), which decomposes cognitive processes into fast, unconscious behavior (System 1) and slow, conscious behavior (System 2). This separation reflects our policy decomposition into a macro and micro part. Other related work in neuroscience and cognitive science include hierarchical models of learning by imitation (Byrne and Russon [2]).

## 3 Long-Term Trajectory Planning

We are interested in learning policies that can produce high quality trajectories, where quality is some global measure of the trajectory (e.g., realistic trajectories as in Figure 1). We first set notation:

- At time $t$, an agent $i$ is in state $s_t^i \in \mathcal{S}$ and takes action $a_t^i \in \mathcal{A}$. The full state and action are $s_t = \left\{s_t^i\right\}_{\text{players } i}$, $a_t = \left\{a_t^i\right\}_{\text{players } i}$. The history of events is $h_t = \left\{(s_u, a_u)\right\}_{0 \le u < t}$.
- Macro policies also use a goal space $\mathcal{G}$, e.g. regions in the court that a player should reach.

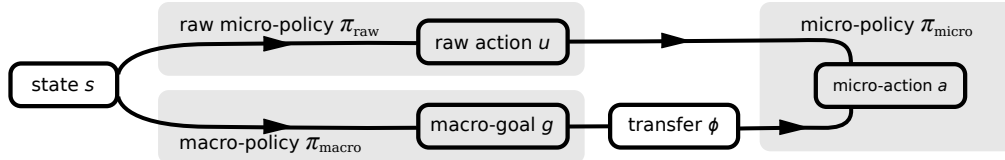

Figure 3: *The general structure of a 2-level hierarchical policy that consists of 1) a raw micro-policy $\pi_{\mathrm{raw}}$ 2) a macro-policy $\pi_{\mathrm{macro}}$ and 3) a transfer function $\phi$. For clarity, we suppressed the indices $i, t$ in the image. The raw micro-policy learns optimal short-term policies, while the macro-policy is optimized to achieve long-term rewards. The macro-policy outputs a macro-goal $g_t^i = \pi_{\mathrm{macro}}(s_t^i, h_t^i)$, which guides the raw micro-policy $u_t^i = \pi_{\mathrm{raw}}(s_t^i, h_t^i)$ in order for the hierarchical policy $\pi_{\mathrm{micro}}$ to achieve a long-term goal $g_t^i$. The hierarchical policy $\pi_{\mathrm{micro}} = \psi(u_t^i, \phi(g_t^i))$ uses a transfer function $\phi$ and synthesis functon $\psi$, see (3) and Section 4.*

- Let $\pi(s_t, h_t)$ denote a policy that maps state and history to a distribution over actions $P(a_t|s_t, h_t)$. If $\pi$ is deterministic, the distribution is peaked around a specific action. We also abuse notation to sometimes refer to $\pi$ as deterministically taking the most probable action $\pi(s_t, h_t) = \mathrm{argmax}_{a \in \mathcal{A}} P(a|s_t, h_t)$ – this usage should be clear from context.

Our main research question is how to design a policy class that can capture the salient properties of how expert agents execute trajectories. In particular, we present a general policy class that utilizes a goal space $\mathcal{G}$ to guide its actions to create such trajectory histories. We show in Section 4 how to instantiate this policy class as a hierarchical network that uses an attention mechanism to combine macro-goals and micro-actions. In our case study on modeling basketball behavior (Section 5.1), we train such a policy to imitate expert demonstrations using a large dataset of tracked basketball games.

## 3.1 Incorporating Macro-Goals

Our main modeling assumption is that a policy should *simultaneously optimize behavior hierarchically on multiple well-separated timescales*. We consider two distinct timescales (*macro* and *micro*-level), although our approach could in principle be generalized to even more timescales. During an episode $[t_0, t_1]$, an agent $i$ executes a sequence of micro-actions $(a_t^i)_{t \geq 0}$ that leads to a macro-goal $g_t^i \in \mathcal{G}$. We do not assume that the start and end times of an episode are fixed. For instance, macro-goals can change before they are reached. We finally assume that macro-goals are relatively static on the timescale of the micro-actions, that is: $dg_t^i/dt \ll 1$.

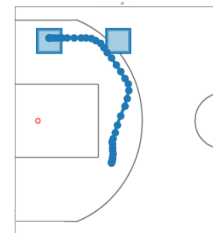

Figure 2: *Depicting two macro-goals (blue boxes) as an agent moves to the top left.*

Figure 2 depicts an example of an agent with two unique macro-goals over a 50-frame trajectory. At every timestep $t$, the agent executes a micro-action $a_t^i$, while the macro-goals $g_t^i$ change more slowly.

We model the interaction between a micro-action $a_t^i$ and a macro-goal $g_t^i$ through a raw micro-action $u_t^i \in \mathcal{A}$ that is independent of the macro-goal. The micro-policy is then defined as:

$$a_t^i = \pi_{\mathrm{micro}}(s_t, h_t) = \mathrm{argmax}_a P^{\mathrm{micro}}(a|s_t, h_t) \tag{1}$$

$$P^{\mathrm{micro}}(a_t^i|s_t, h_t) = \int du dg P(a_t^i|u, g, s_t, h_t) P(u, g|s_t, h_t). \tag{2}$$

Here, we model the conditional distribution $P(a_t^i|u, g, s_t, h_t)$ as a non-linear function of $u, g$:

$$P(a_t^i|u_t^i, g_t^i, s_t, h_t) = \psi(u_t^i, \phi(g_t^i)), \tag{3}$$

where $\phi, \psi$ are transfer and synthesis functions respectively that we make explicit in Section 4. Note that (3) does not explicitly depend on $s_t, h_t$: although it is straightforward to generalize, this did not make a significant difference in our experiments. This decomposition is shown in Figure 3 and can be generalized to multiple scales $l$ using multiple macro-goals $g^l$ and transfer functions $\phi^l$.

## 4 Hierarchical Policy Network

Figure 3 depicts a high-level overview of our hierarchical policy class for generating long-term spatiotemporal trajectories. Both the raw micro-policy and macro-policy are instantiated as recurrent

convolutional neural networks, and the raw action and macro-goals are combined via an attention mechanism, which we elaborate on below.

**Discretization and deep neural architecture.** In general, when using continuous latent variables $g$, learning the model (1) is intractable, and one must resort to approximation methods. We choose to discretize the state-action and latent spaces. In the basketball setting, a state $s_t^i \in \mathcal{S}$ is naturally represented as a 1-hot occupancy vector of the basketball court. We then pose goal states $g_t^i$ as sub-regions of the court that $i$ wants to reach, defined at a coarser resolution than $\mathcal{S}$. As such, we instantiate the macro and micro-policies as convolutional recurrent neural networks, which can capture both predictive spatial patterns and non-Markovian temporal dynamics.

**Attention mechanism for integrating macro-goals and micro-actions.** We model (3) as an attention, i.e. $\phi$ computes a softmax density $\phi(g_t^i)$, over the *output* action space $\mathcal{A}$ and $\psi$ is an element-wise (Hadamard) product. Suppressing indices $i, t$ and $s, h$ for clarity, the distribution (3) becomes

$$\phi_k(g) = \frac{\exp h_\phi(g)_k}{\sum_j \exp h_\phi(g)_j}, \quad P(a_k|u, g) \propto P^{\text{raw}}(u_k|s, h) \cdot \phi_k(g), \quad k = 1 \ldots |\mathcal{A}|, \qquad (4)$$

where $h_\phi(g)$ is computed by a neural network that takes $P^{\text{macro}}(g)$ as an input. Intuitively, this structure captures the trade-off between the macro- and raw micro-policy. On the one hand, the raw micro-policy $\pi_{\text{raw}}$ aims for short-term optimality. On the other hand, the macro-policy $\pi_{\text{macro}}$ can attend via $\phi$ to sequences of actions that lead to a macro-goal and bias the agent towards good long-term behavior. The difference between $u$ and $\phi(g)$ thus reflects the trade-off that the hierarchical policy learns between actions that are good for either short-term or long-term goals.

**Multi-stage learning.** Given a set $D$ of sequences of state-action tuples $(s_t, \hat{a}_t)$, where the $\hat{a}_t$ are 1-hot labels (omitting the index $i$ for clarity), the hierarchical policy network can be trained via

$$\theta^* = \operatorname*{argmin}_\theta \sum_D \sum_{t=1}^T L_t(s_t, h_t, \hat{a}_t; \theta). \qquad (5)$$

Given the hierarchical structure of our model class, we decompose the loss $L_t$ (omitting the index $t$):

$$L(s, h, \hat{a}; \theta) = L_{\text{macro}}(s, h, g; \theta) + L_{\text{micro}}(s, h, \hat{a}; \theta) + R(\theta), \qquad (6)$$

$$L_{\text{micro}}(s, h, \hat{a}; \theta) = \sum_{k=1}^A \hat{a}_k \log \left[ P^{\text{raw}}(u_k|s, h; \theta) \cdot \phi_k(g; \theta) \right], \qquad (7)$$

where $R_t(\theta)$ regularizes the model weights $\theta$ and $k$ indexes $A$ discrete action-values. Although we have ground truths $\hat{a}_t$ for the observable micro-actions, in general we may not have labels for the macro-goals $g_t$ that induce optimal long-term planning. As such, one would have to appeal to separate solution methods to compute the posterior $P(g_t|s_t, h_t)$ which minimizes $L_{t,\text{macro}}(s_t, h_t, g_t; \theta)$.

To reduce complexity and given the non-convexity of (7), we instead follow a multi-stage learning approach with a set of *weak labels* $\hat{g}_t, \hat{\phi}_t$ for the macro-goals $g_t$ and attention masks $\phi_t = \phi(g_t)$. We assume access to such weak labels and only use them in the initial training phases. Here, we first train the raw micro-policy, macro-policy and attention individually, freezing the other parts of the network. The policies $\pi_{\text{micro}}, \pi_{\text{macro}}$ and attention $\phi$ can be trained using standard cross-entropy minimization with the labels $\hat{a}_t, \hat{g}_t$ and $\hat{\phi}_t$, respectively. In the final stage we fine-tune the entire network on objective (5), using only $L_{t,\text{micro}}$ and $R$. We found this approach capable of finding a good initialization for fine-tuning and generating high-quality long-term trajectories.[1] Another advantage of this approach is that the network can be trained using gradient descent during all stages.

## 5 Case Study on Modeling Basketball Behavior

We applied our approach to modeling basketball behavior data. In particular, we focus on imitating the players' movements, which is a challenging problem in the spatiotemporal planning setting.

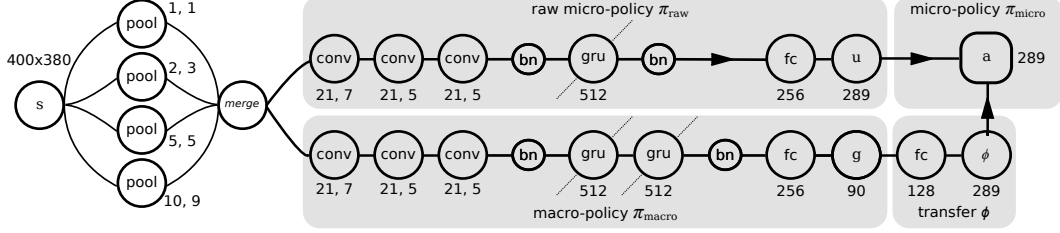

*Figure 4: Network architecture and hyperparameters of the hierarchical policy network. For clarity, we suppressed the indices $i, t$ in the image. Max-pooling layers (numbers indicate kernel size) with unit stride upsample the sparse tracking data $s_t$. The policies $\pi_{\text{raw}}, \pi_{\text{macro}}$ use a convolutional (kernel size, stride) and GRU memory (number of cells) stack to predict $u_t^i$ and $g_t^i$. Batch-normalization "bn" (Ioffe and Szegedy [9]) is applied to stabilize training. The output attention $\phi$ is implemented by 2 fully-connected layers (number of output units). Finally, the network predicts the final output $\pi_{\text{micro}}(s_t, h_t) = \pi_{\text{raw}}(s_t, h_t) \odot \phi(g_t^i)$.*

## 5.1 Experimental Setup

We validated the hierarchical policy network (HPN) by learning a movement policy of individual basketball players that predicts as the micro-action the *instantaneous velocity* $v_t^i = \pi_{\text{micro}}(s_t, h_t)$.

**Training data.** We trained the HPN on a large dataset of tracking data from professional basketball games (Yue et al. [16]). The dataset consists of *possessions* of variable length: each possession is a sequence of tracking coordinates $s_t^i = (x_t^i, y_t^i)$ for each player $i$, recorded at 25 Hz, where one team has continuous possession of the ball. Since possessions last between 50 and 300 frames, we sub-sampled every 4 frames and used a fixed input sequence length of 50 to make training feasible. Spatially, we discretized the left half court using $400 \times 380$ cells of size 0.25ft $\times$ 0.25ft. For simplicity, we modeled every player identically using a single policy network. The resulting input data for each possession is grouped into 4 channels: the ball, the player's location, his teammates, and the opposing team. After this pre-processing, we extracted 130,000 tracks for training and 13,000 as a holdout set.

**Labels.** We extracted micro-action labels $\hat{v}_t^i = s_{t+1}^i - s_t^i$ as 1-hot vectors in a grid of $17 \times 17$ unit cells. Additionally, we constructed a set of weak macro-labels $\hat{g}_t, \hat{\phi}_t$ by heuristically segmenting each track using its stationary points. The labels $\hat{g}_t$ were defined as the next stationary point. For $\hat{\phi}_t$, we used 1-hot velocity vectors $v_{t,\text{straight}}^i$ along the straight path from the player's location $s_t^i$ to the macro-goal $g_t^i$. We refer to the supplementary material for additional details.

**Model hyperparameters.** To generate smooth rollouts while sub-sampling every 4 frames, we simultaneously predicted the next 4 micro-actions $a_t, \ldots, a_{t+3}$. A more general approach would model the dependency between look-ahead predictions as well, e.g. $P(\pi_{t+\Delta+1}|\pi_{t+\Delta})$. However, we found that this variation did not outperform baseline models. We selected a network architecture to balance performance and feasible training-time. The macro and micro-policy use GRU memory cells Chung et al. [3] and a memory-less 2-layer fully-connected network as the transfer function $\phi$, as depicted in Figure 4. We refer to the supplementary material for more details.

**Baselines.** We compared the HPN against two natural baselines.

1. A policy with only a raw micro-policy and memory (GRU-CNN) and without memory (CNN).
2. A hierarchical policy network H-GRU-CNN-CC without an attention mechanism, which instead learns the final output from a concatenation of the raw micro- and macro-policy.

**Rollout evaluation.** To evaluate the quality of our model, we generated rollouts $(s_t; h_{0,r_0})$ with *burn-in period $r_0$*, These are generated by 1) feeding a ground truth sequence of states $h_{0,r_0} = (s_0, \ldots, s_{r_0})$ to the policy network and 2) for $t > r_0$, predicting $a_t$ as the mode of the network output (1) and updating the game-state $s_t \rightarrow s_{t+1}$, using ground truth locations for the other agents.

## 5.2 How Realistic are the Generated Trajectories?

The most holistic way to evaluate the trajectory rollouts is via visual analysis. Simply put, would a basketball expert find the rollouts by HPN more realistic than those by the baselines? We begin by first visually analyzing some rollouts, and then present our human preference study results.

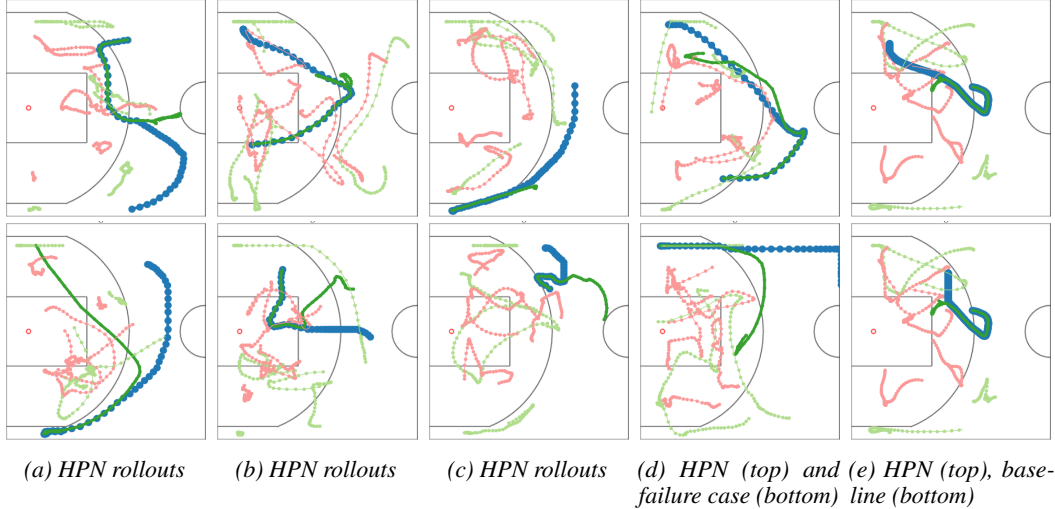

| (a) HPN rollouts | (b) HPN rollouts | (c) HPN rollouts | (d) HPN (top) and failure case (bottom) | (e) HPN (top), baseline (bottom) |

*Figure 5: Rollouts generated by the HPN (columns a, b, c, d) and baselines (column e). Each frame shows an offensive player (dark green), a rollout (blue) track that extrapolates after 20 frames, the offensive team (light green) and defenders (red). Note we do not show the ball, as we did not use semantic basketball features (i.e "currently has the ball") during training. The HPN rollouts do not memorize training tracks (column a) and display a variety of natural behavior, such as curving, moving towards macro-goals and making sharp turns (c, bottom). We also show a failure case (d, bottom), where the HPN behaves unnaturally by moving along a straight line off the right side of the court – which may be fixable by adding semantic game state information. For comparison, a hierarchical baseline without an attention model, produces a straight-line rollout (column e, bottom), whereas the HPN produces a more natural movement curve (column e, top).*

| Model comparison | Experts | | Non-Experts | | All | |
|---|---|---|---|---|---|---|
| | W/T/L | Avg Gain | W/T/L | Avg Gain | W/T/L | Avg Gain |
| VS-CNN | 21 / 0 / 4 | 0.68 | 15 / 9 / 1 | 0.56 | 21 / 0 / 4 | 0.68 |
| VS-GRU-CNN | 21 / 0 / 4 | 0.68 | 18 / 2 / 5 | 0.52 | 21 / 0 / 4 | 0.68 |
| VS-H-GRU-CNN-CC | 22 / 0 / 3 | 0.76 | 21 / 0 / 4 | 0.68 | 21 / 0 / 4 | 0.68 |
| VS-GROUND TRUTH | 11 / 0 / 14 | -0.12 | 10 / 4 / 11 | -0.04 | 11 / 0 / 14 | -0.12 |

*Table 1: Preference study results. We asked basketball experts and knowledgeable non-experts to judge the relative quality of policy rollouts. We compare HPN with ground truth and 3 baselines: a memory-less (CNN ) and memory-full (GRU-CNN ) micro-policy and a hierarchical policy without attention (GRU-CNN -CC). For each of 25 test cases, HPN wins if more judges preferred the HPN rollout over a competitor. Average gain is the average signed vote (1 for always preferring HPN , and -1 for never preferring). We see that the HPN is preferred over all baselines (all results against baselines are significant at the 95% confidence level). Moreover, HPN is competitive with ground truth, indicating that HPN generates realistic trajectories within our rollout setting. Please see the supplementary material for more details.*

**Visualization.** Figure 5 depicts example rollouts for our HPN approach and one example rollout for a baseline. Every rollout consists of two parts: 1) an initial ground truth phase from the holdout set and 2) a continuation by either the HPN or ground truth. We note a few salient properties. First, the HPN does not memorize tracks, as the rollouts differ from the tracks it has trained on. Second, the HPN generates rollouts with a high dynamic range, e.g. they have realistic curves, sudden changes of directions and move over long distances across the court towards macro-goals. For instance, HPN tracks do not move towards macro-goals in unrealistic straight lines, but often take a curved route, indicating that the policy balances moving towards macro-goals with short-term responses to the current state (see also Figure 6b). In contrast, the baseline model often generates more constrained behavior, such as moving in straight lines or remaining stationary for long periods of time.

**Human preference study.** Our primary empirical result is a preference study eliciting judgments on the relative quality of rollout trajectories between HPN and baselines or ground truth. We recruited seven experts (professional sports analysts) and eight knowledgeable non-experts (e.g., college basketball players) as judges.

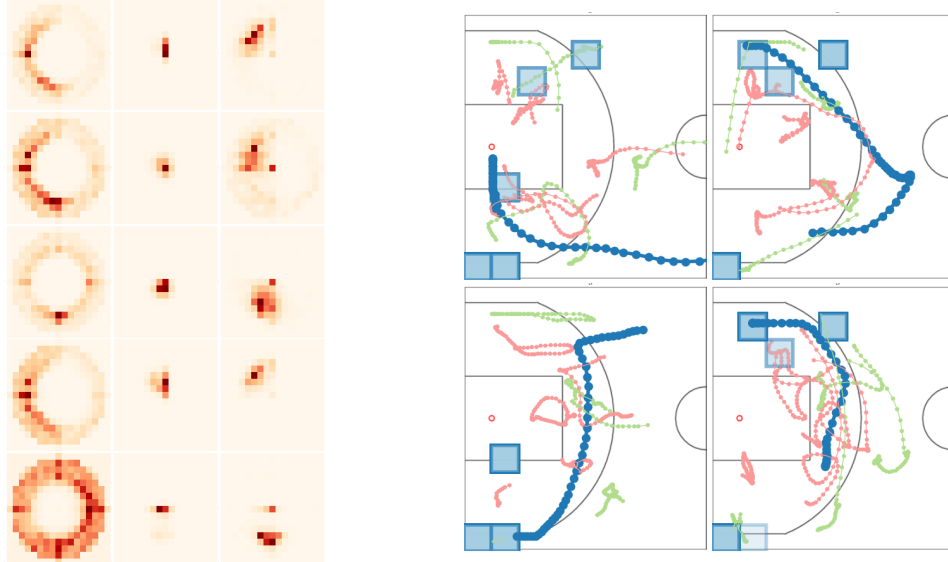

*(a) Predicted distributions for attention masks $\phi(g)$ (left column), velocity (micro-action) $\pi_{\mathrm{micro}}$ (middle) and weighted velocity $\phi(g) \odot \pi_{\mathrm{micro}}$ (right) for basketball players. The center corresponds to 0 velocity.*

*(b) Rollout tracks and predicted macro-goals $g_t$ (blue boxes). The HPN starts the rollout after 20 frames. Macro-goal box intensity corresponds to relative prediction frequency during the trajectory.*

*Figure 6: **Left:** Visualizing how the attention mask generated from the macro-policy interacts with the micro-policy $\pi_{\mathrm{micro}}$. Row 1 and 2: the micro-policy $\pi_{\mathrm{micro}}$ decides to stay stationary, but the attention $\phi$ goes to the left. The weighted result $\phi \odot \pi_{\mathrm{micro}}$ is to move to the left, with a magnitude that is the average. Row 3) $\pi_{\mathrm{micro}}$ wants to go straight down, while $\phi$ boosts the velocity so the agent bends to the bottom-left. Row 4) $\pi_{\mathrm{micro}}$ goes straight up, while $\phi$ goes left: the agent goes to the top-left. Row 5) $\pi_{\mathrm{micro}}$ remains stationary, but $\phi$ prefers to move in any direction. As a result, the agent moves down. **Right:** The HPN dynamically predicts macro-goals and guides the micro-policy in order to reach them. The macro-goal predictions are stable over a large number of timesteps. The HPN sometimes predicts inconsistent macro-goals. For instance, in the bottom right frame, the agent moves to the top-left, but still predicts the macro-goal to be in the bottom-left sometimes.*

Because all the learned policies perform better with a "burn-in" period, we first animated with the ground truth for 20 frames (after subsampling), and then extrapolated with a policy for 30 frames. During extrapolation, the other nine players do not animate.[2] For each test case, the judges were shown an animation of two rollout extrapolations of a specific player's movement: one generated by the HPN, the other by a baseline or ground truth. The judges then chose which rollout looked more realistic. Please see the supplementary material for details of the study.

Table 1 shows the preference study results. We tested 25 scenarios (some corresponding to scenarios in Figure 6b). HPN won the vast majority of comparisons against the baselines using expert judges, with slightly weaker but still very positive results using non-expert judgments. HPN was also competitive with ground truth. These results suggest that HPN can generate high-quality player trajectories that are significant improvements over baselines, and approach the ground truth quality in this comparison setting.

## 5.3 Analyzing Macro- and Micro-policy Integration

Our model integrates the macro- and micro-policy by converting the macro-goal into an attention mask on the micro-action output space, which intuitively guides the micro-policy towards the macro-goal. We now inspect our macro-policy and attention mechanism to verify this behavior.

Figure 6a depicts how the macro-policy $\pi_{\mathrm{macro}}$ guides the micro-policy $\pi_{\mathrm{micro}}$ through the attention $\phi$, by attending to the direction in which the agent can reach the predicted macro-goal. The attention model and micro-policy differ in semantic behavior: the attention favors a wider range of velocities and larger magnitudes, while the micro-policy favors smaller velocities.

| Model | $\Delta = 0$ | $\Delta = 1$ | $\Delta = 2$ | $\Delta = 3$ | Macro-goals $g$ | Attention $\phi$ |
|---|---|---|---|---|---|---|
| CNN | 21.8% | 21.5% | 21.7% | 21.5% | - | - |
| GRU-CNN | 25.8% | 25.0% | 24.9% | 24.4% | - | - |
| H-GRU-CNN-CC | 31.5% | 29.9% | 29.5% | 29.1% | 10.1% | - |
| H-GRU-CNN-STACK | 26.9% | 25.7% | 25.9% | 24.9% | 9.8% | - |
| H-GRU-CNN-ATT | **33.7%** | **31.6%** | **31.0%** | **30.5%** | 10.5% | - |
| H-GRU-CNN-AUX | 31.6% | 30.7% | 29.4% | 28.0% | 10.8% | 19.2% |

*Table 2: Benchmark Evaluations. $\Delta$-step look-ahead prediction accuracy for micro-actions $a^i_{t+\Delta} = \pi(s_t)$ on a holdout set, with $\Delta = 0, 1, 2, 3$. H-GRU-CNN-STACK is an HPN where predictions are organized in a feed-forward stack, with $\pi(s_t)_t$ feeding into $\pi(s_t)_{t+1}$. Using attention (H-GRU-CNN-ATT) improves on all baselines in micro-action prediction. All hierarchical models are pre-trained, but not fine-tuned, on macro-goals $\hat{g}$. We report prediction accuracy on the weak labels $\hat{g}, \hat{\phi}$ for hierarchical models. H-GRU-CNN-AUX is an HPN that was trained using $\hat{\phi}$. As $\hat{\phi}$ optimizes for optimal long-term behavior, this lowers the micro-action accuracy.*

Figure 6b depicts predicted macro-goals by HPN along with rollout tracks. In general, we see that the rollouts are guided towards the predicted macro-goals. However, we also observe that the HPN makes some inconsistent macro-goal predictions, which suggests there is still room for improvement.

## 5.4 Benchmark Analysis

We finally evaluated using a number of benchmark experiments, with results shown in Table 2. These experiments measure quantities that are related to overall quality, albeit not holistically. We first evaluated the 4-step look-ahead accuracy of the HPN for micro-actions $a^i_{t+\Delta}, \Delta = 0, 1, 2, 3$. On this benchmark, the HPN outperforms all baseline policy networks when not using weak labels $\hat{\phi}$ to train the attention mechanism, which suggests that using a hierarchical model can noticeably improve the short-term prediction accuracy over non-hierarchical baselines.

We also report the prediction accuracy on weak labels $\hat{g}, \hat{\phi}$, although they were only used during pre-training, and we did not fine-tune for accuracy on them. Using weak labels one can tune the network for both long-term and short-term planning, whereas all non-hierarchical baselines are optimized for short-term planning only. Including the weak labels $\hat{\phi}$ can lower the accuracy on short-term prediction, but increases the quality of the on-policy rollouts. This trade-off can be empirically set by tuning the number of weak labels used during pre-training.

## 6 Limitations and Future Work

We have presented a hierarchical memory network for generating long-term spatiotemporal trajectories. Our approach simultaneously models macro-goals and micro-actions and integrates them using a novel attention mechanism. We demonstrated significant improvement over non-hierarchical baselines in a case study on modeling basketball player behavior.

There are several notable limitations to our HPN model. First, we did not consider all aspects of basketball gameplay, such as passing and shooting. We also modeled all players using a single policy whereas in reality player behaviors vary (although the variability can be low-dimensional (Yue et al. [16])). We only modeled offensive players: an interesting direction is modeling defensive players and integrating adversarial reinforcement learning (Panait and Luke [13]) into our approach. These issues limited the scope of our preference study, and it would be interesting to consider extended settings.

In order to focus on the HPN model class, we only used the imitation learning setting. More broadly, many planning problems require collecting training data via exploration (Mnih et al. [11]), which can be more challenging. One interesting scenario is having two adversarial policies learn to be strategic against each other through repeatedly game-play in a basketball simulator. Furthermore, in general it can be difficult to acquire the appropriate weak labels to initialize the macro-policy training.

From a technical perspective, using attention in the output space may be applicable to other architectures. More sophisticated applications may require multiple levels of output attention masking.

**Acknowledgments.** This research was supported in part by NSF Award #1564330, and a GPU donation (Tesla K40 and Titan X) by NVIDIA.

## Footnotes

[1] As $u_t$ and $\phi(g_t)$ enter symmetrically into the objective (7), it is hypothetically possible that the network converges to a symmetric phase where the predictions $u_t$ and $\phi(g_t)$ become identical along the entire trajectory. However, our experiments suggest that our multi-stage learning approach separates timescales well between the micro- and macro policy and prevents the network from settling in such a redundant symmetric phase.

[2]We chose this preference study design to focus the qualitative comparison on the plausibility of individual movements (e.g. how players might practice alone), as opposed to strategically coordinated team movements.

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
