[Supplementary Material]

# Generating Long-term Trajectories Using Deep Hierarchical Networks

**Supplementary Material**

**Stephan Zheng**
Caltech
stzheng@caltech.edu

**Yisong Yue**
Caltech
yyue@caltech.edu

**Patrick Lucey**
STATS
plucey@stats.com

## 1 Data format and pre-processing

In the temporal domain, the raw tracking data is recorded at 25 Hz. We sub-sampled the possessions in the temporal domain by extracting sequences of 200 frames (= 8 seconds) long at random starting points, using only every 4 frames to reduce the model complexity and redundancy. In this way, we obtained 130.000 training tracks of 50 frames and 13.000 tracks as a holdout set.

In the spatial domain, without loss of generality, we only considered possessions in a half-court of size $50 \times 45$ feet, by flipping all coordinates if needed. We used a discretized representation of the spatial domain and sampled the raw input tracking data at different spatial resolutions at train-time: this is necessary as the raw tracking data is very sparse in the natural 1-hot occupancy representation using a 1-ft by 1-ft spatial resolution (= $50 \times 45 = 2250$ grid cells). For training, we used a resolution of 0.25ft $\times$ 0.25ft (= $400 \times 380$ cells). We found that dynamically downsampling the input data at multiple resolutions ($400 \times 360$, $200 \times 180$, $50 \times 45$, $25 \times 15$, $10 \times 9$) using max-pooling layers provided enough signal for the model to be trainable.

As the policy network is agnostic of the individual player identities, we grouped the input data for each possession into 4 separate channels: the ball, the tracked player, his teammates and the opposing team. At train-time, the input sequences are given to the model as a 5d-tensor in the format $(batch, timestep, entity, x, y)$.

**Micro-action labels.** We extracted micro-goal labels by computing the cell velocity of the discretized input tracking data. We used a velocity grid of $17 \times 17$ cells, corresponding to velocities in the range of $(-1, 1)$ feet per 1/25s in both directions. Velocities outside this range were clipped, which applied to <1% of all frames.

**Macro-goal labels.** To extract macro-goals, we used a heuristic segmentation of the player tracks, where each segment is bounded by periods of relative stationary behavior. We used a threshold of 0.25 ft per 1/25s to determine these stationary points. We additionally imposed a minimal segment length of 5 frames, to avoid redundant segmentations.

The macro-goals were then defined as the cluster in which the player was located during the stationary periods. We used a clustering of $10 \times 9$ boxes of $5 \times 5$ft of the basketball court. The final macro-goal is always the final position of the track in the possession.

**Straight-line micro-action labels.** We also extracted the straight-line velocities between the player location $s_t$ and the macro-goal $g_t$. The magnitude of the velocity was chosen randomly between 1 and 7 unit cells, with the direction fixed by the vector $s_t - g_t$. We stored these velocity labels as attention mask labels, using the same $17 \times 17$ velocity grid definition as for the micro-action labels. We used these weak labels to pre-train the attention mechanism $m$ through standard cross-entropy minimization.

**Training procedure.** Since our input data is extremely sparse in the occupancy representation, to make training feasible we upsampled the input to multiple resolutions by applying max-pooling layers after the input.

During all stages, we used RMSprop with momentum and a batch-size of 16. We performed a search over training hyperparameters and found that a learning-rate of $10^{-3}$ (pre-training), $10^{-5} - 10^{-6}$ (fine-tuning), decay rate $10^{-4}$ and momentum-parameter 0.9 gave good results.

**Regularization.** To improve the robustness of the policy rollouts we distorted the input tracks with uniform random translations of <8 unit cells, used $L_2$-regularization on the attention masks $\phi$ and micro-actions $a$, and injected Gaussian noise with $\sigma = 10^{-3}$ after the convolutional layers. Moreover, we used batch-normalization [1] and gradient clipping to stabilize training.

## 2 Additional rollout visualizations

We present some additional rollout visualizations of the baseline models in Figures (1), (2), (3).

*Figure 1: Rollouts generated by baseline CNN models. Each frame shows an offensive player (dark green), the ball (blue), the offensive team (light green) and defenders (red). Top row: ground truth. Bottom row: a model rollout (dark green) that starts after having seen 20 frames (after subsampling) of the ground truth. The most prominent failure case of this baseline is that the agent remains stationary, as in the bottom row, the agent stays largely in the same position when the model rollout starts.*

*Figure 2: Rollouts generated by baseline GRU-CNN models. Each frame shows an offensive player (dark green), the ball (blue), the offensive team (light green) and defenders (red). Top row: ground truth. Bottom row: a model rollout (dark green) that start after having seen 20 frames (after subsampling) of the ground truth. The rollouts demonstrate several prominent failure cases: the policy (dark green) remains stationary (when the model rollout starts), or often follows straight horizontal trajectories.*

*Figure 3: Rollouts generated by baseline H-GRU-CNN-CC models. Each frame shows an offensive player (dark green), the ball (blue), the offensive team (light green) and defenders (red). Top row: ground truth. Bottom row: a model rollout (dark green) that start after having seen 20 frames (after subsampling) of the ground truth. Although in some cases the model display plausible behavior (1st column), a prominent failure case of the model is to follow unrealistic straight trajectories in unrealistic directions.*

## 3   User study

**Setup.** To test whether our model produces realistic basketball play behavior, we performed a small-scale user study with domain experts. In the study, the subjects were presented with a set of possessions and rollouts through a web interface, see Figure 4. Broadly speaking, there are two qualities one could evaluate: the plausibility of (1) individual movements (e.g. how players might practice alone) versus (2) strategically coordinated team movements. We designed our user study to focus on (1) and minimize conflation with (2). All models perform pretty poorly w.r.t. (2).

For each possession, the ground truth movements of all players were shown for 20 frames. After these initial frames, a single player continued to move for 30 frames, while the other players were frozen. The subjects were presented with two alternatives for the continuation of the player's movement: one generated by a rollout of the hierarchical policy network, the other either by 1) a baseline or 2) the ground truth. The subjects were then asked to indicate which alternative looked more realistic.

We used a pool of 7 domain experts (professional sports data analysts) and 8 basketball enthusiasts, and 25 possessions as our test data. Each participant was presented with 100 test cases, consisting of 4 pair-wise comparisons of the hierarchical policy network rollout with one of the 3 baselines or the ground truth rollout, for each possession.

*Figure 4: User study interface for human judgements on the realism of the tracks generated by the hierarchical policy network.*

# References

[1] Sergey Ioffe and Christian Szegedy. Batch Normalization: Accelerating Deep Network Training by Reducing Internal Covariate Shift. pages 448–456, 2015.