[Reviews · NeurIPS 2016]

Reviewer 1

Summary

This paper presents an approach to learn hierarchical policy models from demonstration, and demonstrates the proposed approach on generating player behavior in basketball. The authors use deep neural networks combined with an attention mechanism for combining across high level objectives/goals and micro-actions.

Qualitative Assessment

The aim of this work is very interesting. However, in its current form, it is hard to understand and assess the proposed technical contribution, especially for readers such as myself who are familiar/researchers in RL but do not have deep knowledge of deep learning. In particular, I found the main section on the proposed method, section 4, very hard to parse as a non-expert in deep learning. Some parts were not defined (for example, what is L in Equation 5? Loss? How is loss defined here?). The work also didn't seem to be sufficiently situated in the literature, or compared to alternate approaches. For example, both George Konidaris and Sanjoy Krishnan seem to have relevant work on learning skills or macro-actions or policies from demonstration, but these were not compared to nor sited. I highly encourage the authors to revise and submit again. I highly encourage the authors to revise and submit again. Related work of Sanjoy Krishnan or George Konidaris G.D. Konidaris, S.R. Kuindersma, R.A. Grupen and A.G. Barto. Robot Learning from Dem\ onstration by Constructing Skill Trees. The International Journal of Robotics Resear\ ch 31(3), pages 360-375, March 2012. Niekum, Scott, et al. "Learning and generalization of complex tasks from unstructure\ d demonstrations." 2012 IEEE/RSJ International Conference on Intelligent Robots and \ Systems. IEEE, 2012. Han, Weiqiao, Sergey Levine, and Pieter Abbeel. "Learning compound multi-step contro\ llers under unknown dynamics." Intelligent Robots and Systems (IROS), 2015 IEEE/RSJ \ International Conference on. IEEE, 2015. Murali, A., Garg, A., Krishnan, S., Pokorny, F. T., Abbeel, P., Darrell, T., & Goldb\ erg, K. (2016, May). TSC-DL: Unsupervised trajectory segmentation of multi-modal sur\ gical demonstrations with Deep Learning. In 2016 IEEE International Conference on Ro\ botics and Automation (ICRA) (pp. 4150-4157). IEEE. - In practice, where would weak labels come from?

Confidence in this Review

1-Less confident (might not have understood significant parts)


Reviewer 2

Summary

The paper presents a hierarchical model for trajectories planning. The authors applied it to basketball (player) trajectory prediction. The gist of the model is to have two policy networks, one called "micro-planner" that computes a distribution on actions a_{micro} ~ \pi(s; \theta_{micro}), and one called "macro-planner" that computes a distribution on "goals" \pi(s; \theta_{macro}). A "goal" g is a sub-region of the court (coarser grain than s) that the player wants to go to. There is then a "transfer function" m that maps g to an attention mask m_g (in the space of actions) that is pointwise multiplied (Hadamard product) with the distribution on actions (a_{micro}). Both the micro- and macro-planners are trained (through cross-entropy loss) with the other parts frozen, with the labeling being ground truth micro-action labels. The experiments include a visual analysis by humans of the trajectory generated by the planner, and a prediction accuracy benchmark of the different ablations of the model.

Qualitative Assessment

This is overall a nice paper, quite easy to read, built on an interesting idea of using an attention mechanism to specify sub-goals, but I think that it has a too many technical limitations for inclusion in NIPS as is. Limitations: - The title feels inappropriate, there is no memory as in "memory networks", the only form of memory in the model comes from the GRUs. (Recurrent and) Hierarchical Trajectory Planning Using ConvNets would be OK. - The multi-stage learning, in which the authors train the micro-planner, macro-planner, and attention mechanism individually "freezing the other parts of the network", vs. end-to-end training with the final prediction m_g(a), back-propagating both through the attention, the micro-planner and macro-planner. Both training methods should at least be benchmarked. - Using a discrete 17x17 action space vs. using a continuous 2D space, or (in particular) continuous or discretized (r, \theta) polar coordinates is not justified in the article. It seems to me that it makes the training harder than it should. A mean square error on a continuous space (even more so with two components on polar coordinates) would extract more information per training sample as a slight error in direction and/or intensity is not the same as going the opposite direction (which is the case for the current loss). - In the experiments, prediction is limited as all the other players stop moving (page 7 line 209, or watch the supplementary material). I wonder how much of an edge this gives to "random" and some other baselines. This is a concern as it makes it harder to draw conclusions from the experiments. - Table 2. lacks some baselines like "random" and "keep moving in the same direction". The results in table 2 seem to show that training the attention mechanism is not always beneficial... Unclear: - page 5 lines 164-165: "we sub-sampled every 4 frames" vs. line 175 "while sub-sampling at 4Hz". 25Hz / 4 = 6.25 Hz. One or the other, I get the idea, but please specify. - precise that "frames" in section 5 (e.g. page 7 line 208) refer to frames after subsampling. - Phrase "95% statistically significant" better.

Confidence in this Review

2-Confident (read it all; understood it all reasonably well)


Reviewer 3

Summary

This paper considers long-term trajectory planning using hierarchical policy models, in order to model the macro-micro behavior. The model is instantiated using deep memory network with attention mechanism, through which the macro-planner guides the micro-planner. It is shown that the proposed method generates more realistic player trajectories.

Qualitative Assessment

In the first paragraph of section 3.1, the notation g_t is not defined before its usage here. Moreover, the goal space is not defined in a concrete manner. Is it a finite dimensional space here? What is the dimension? How is it represented in practical problems? It is not clear how equation (3) is derived as both the left-hand side and the right-hand side have P(a_t | s_t, h_{t-n,t}), which is very strange. How are the weak labels \hat{g}_t and \hat{m}_t obtained?

Confidence in this Review

2-Confident (read it all; understood it all reasonably well)


Reviewer 4

Summary

This paper proposes an algorithm to learn behavior based on long-term goals, with the prediction of player movement in basketball games as application. The core idea is to train a hierarchical model that predicts low-level actions by combining a so-called "micro-planner" (that uses only the current state as input) with a "macro-planner" (that also takes into account the estimated long-term goal). The estimation of the long-term goal requires initial seeding with heuristically-defined targets: for instance in basketball the authors use stationary points (when players stop moving) as such goals. The main experimental evaluation consists in comparing (with human judges) the quality of the completion of basketball player traces, vs. those generated by baselines.

Qualitative Assessment

The idea of learning higher-level goal abstractions to drive agent behavior is definitely a very relevant one, and as far as I am aware the proposed approach to tackling this problem is new. I found however the "Related Work" section to be under-developed, with too few links to previous work in the areas of hierarchical reinforcement learning, goal recognition and trajectory planning. To me, the biggest issue with this submission is an overall lack of clarity. I was really lost my first time reading through, and some points only made sense to me near the end, so I had to re-read it again to actually understand what was going on. I did not realize until eq. 6-7 (p. 4) that it was actually supervised learning: the mentions of planning and reinforcement learning had thrown me off. I guess it can be argued that predicting long-term goals is a form of "planning", however calling the basic state-to-action predictor a micro-"planner" seems inappropriate to me, considering it is only trying to imitate expert trajectories with absolutely no look-ahead into the future. Here are additional details I found unclear: - Notations in general. Which quantities are scalars / vectors / matrices? What is the difference between bold s_t and non-bold s_t l.83? (l.160 really confuses me). Why is m_g a probability on "m" in eq. 2 but a function of "a" in eq. 4? Why is a_t found in two probabilities in eq. 3? (I failed to understand eq. 3 and how it is used in the model) What is the index i l.133? (I guess a player, but please make it clear -- also i appears out of nowhere l.142 although it is not used in the equations above) What does the multiplication by \hat{a}_t in eq. 7 mean? (I believe I understand now but it is confusing) - Why is it important to "assume that macro-goals are relatively static"? - Is eq. 6 used at all? (it is said fine-tuning is done on eq. 7 only) Besides the above clarity issues, I also have some concerns on the overall methodology, that hopefully authors may address in their reply. As far as I can understand the predictor for the attention mask m only takes the predicted macro-goal g as input: it means the prediction will be the same for a given g, regardless of the position of the player! This seems like a significant issue to me, in particular for goals the player may approach from different sides (requiring moving in completely different directions). Also, during extrapolation, since the generated player trajectory is not the same as the one from the seeding sequence, the movements of the ball and other players may make no sense: as a result, the model may eventually be making predictions from very "unrealistic" situations never seen during training, and it is hard to tell how it would behave (in theory it would be possible for a model generating great-looking trajectories on realistic data to completely break in this experimental setup). I realize it is a hard problem and I have no magic answer to it, but I feel like this limitation should at least be mentioned. It would have also been interesting to investigate to which extent the input data beyond the considered player's trace impact the model's output. Other minor points / typos: - l.96 - "should simultaneously optimizes" - l.176 "we used an simultaneously predicted the next 4 micro-actions" - Fig. 5 caption: "which may be fixable by using additional game state information": it is not clear to me why additional game state information would fix this specific situation - Fig. 6 caption: "bottom-right" on the last line should be "bottom-left" - l.222: "We now inspect this our macro-planner" - The 4-line Conclusion is a bit weak, maybe consider merging it with Section 6? - "sub-sampling temporally at 4 Hz": does that mean keeping one frame out of four? That would make it 25/4 Hz Update after author feedback: Thanks for the response. I can indeed see why in the specific case of this basketball dataset, conditioning the mask on the state may not help (but in general I think it would make sense to do it). My overall feeling after reading other reviews and the author feedback is that the algorithm needs to presented in a clearer way, and be better motivated / evaluated against simpler or previously proposed methods. I appreciate the willingness to make the data public (but note that it makes it even more important to have clear quantitative benchmarks other people can compare against).

Confidence in this Review

2-Confident (read it all; understood it all reasonably well)


Reviewer 5

Summary

The paper presents a method for long-term trajectory planning using macro-goals. The macro-goals affect micro-actions through a novel attention mechanism. The method outperforms non-hierarchical approaches on modeling basketball players' behavior.

Qualitative Assessment

The main message of the paper, the idea of using macro-goals to affect the choice of micro-actions, is rather attractive. However, I found it difficult to follow the theory behind the proposed method. Here are my notes concerning Section 3 "Long-Term Trajectory Planning". 1. Line 84. Typo: t should be in half-open interval [p,q), because otherwise formulas (1) and (3) don't make sense, since the action a_t would be included in history h_{t-n,t}. 2. Lines [109,113). It is unclear what m is. Is it a function or a random variable? It is used apparently in both senses. If it is a function, then for a constant goal it will return a sequence of constant states and actions. If m is a random variable, its values at different time steps will be uncorrelated. In both cases, the purpose and use of m remains unclear. 3. Formula (3). Typo? Should the first integrand be P(a | g, s, h) and not P(a | s, h)? Compare to Formula (1). 4. Formula (4). As seen from Formula (2), m_g is a probability density function. What kind of object is m_g(a) then? Since Formula (4) says that it is a Hadamard product of m_g with a, and assuming a is a vector, I guess m_g is also a vector. Is it a conflict of notation? Some clarification is needed here. I also feel that notation \pi(s | g) is somewhat misleading, because \pi(s) = p(a | s), so s is fixed, just like g. I think \pi(s, g) would be perhaps more suitable. Although the level of technical details was insufficient to fully appreciate the inner workings of the proposed approach, I think the ideas presented in the paper deserve attention and provide a novel view on how to combine macro-goals along with micro-actions using memory networks. One could argue that experiments lack objective measure of performance, relying only on subjective visual impressions of experts, but on the other hand, it is indeed hard to say which basketball trajectories are good and which are bad.

Confidence in this Review

1-Less confident (might not have understood significant parts)


Reviewer 6

Summary

The paper describes a specific instantiation of a deep memory network with an attention mechanism which was designed for hierarchical, long-term, action planing in complicated problems. The network is trained via behavioral cloning of expert trajectories and evaluated on a novel task: predicting basketball player behavior from captured real play-data. The different choices of the model architecture are studied in detail and thoroughly evaluated.

Qualitative Assessment

The paper is well written in general (aside from the minor stylistic issues mentioned below) and addresses an important problem: That of learning hierarchical behavioral policies from data. The chosen benchmark is interesting and is a welcome change from the standard benchmarks that are often used for control tasks/imitation learning. The experiments presented in the paper are thorough and different model choices are studied in separation, allowing the authors to attribute the performance improvement of their model to concrete decisions made in the modeling. This is a big plus. The model formulation seems reasonable and the connection between hierarchical policies and draws from attention mechanisms for language modeling (a very reasonable idea in my mind). Aside from these clearly positive points the paper unfortunately also has a few issues. I will name them and then detail my criticism in turn. Overall, in my eyes, the paper could still well be accepted if the issues with the slightly confusing notation and the formulation of the loss function/training details are addressed (and the bad quality figures should obviously be replaced). Related work details and citations: - You fail to cite related work from imitation learning/inverse reinforcement learning (IRL)/apprenticeship learning that could be used to solve the presented problem. In fact the presented algorithm can be interpreted as one instance of imitation learning to which the community refers to with the term "behavioral cloning". - In the related work you interchangeably use policy and planner, which at least to someone from the RL community seems somewhat odd. It also seems that calling DQN a "shallow planner" is a bit unfortunate terminology :). Notation issues/Loss functions used: - In Equation (2) and (3) the auxiliary transfer function is is not conditioned on s_t but from Figure 3 and your network definition further below it seems that, obviously g is predicted from s_t and thus m_g should be conditional on it. Do you omit this simply because you assume it is an independent random event ? - Am I correct in assuming that the actual posterior derived in Equation (3) (and for the computation of which you discretize the state and action space) is never used during training ? It seems each part of the model is trained in a supervised manner and then fine tuning only occurs on the micro action level without computing the integral. Or does the probability density to which you refer in Equation (7) actually correspond to Equation (3) ? This is unclear from the context. - It also seems that suddenly between Equation (3) and Equation (7) the dependency on the history disappears. I presume this is just an error, but it is sloppy notation wise nonetheless. - My MAIN ISSUE with the loss function is captured by the footnote you have on page 4. Although it is not completely obvious what the loss in Equation (7) is (see comment above) it indeed seems that it is symmetrical in the output of the micro and macro networks. Given that these operate on the same input, and there is no supervision on the macro-goals during fine-tuning, it seems that the objective function then has a trivial local minimum in which one of the networks just outputs a uniform distribution (unless you actually correctly integrate as in Equation (3) in which case I am not 100% sure about this). If this is the case then the only thing preventing the model from converging to this solution is the initialization and the fact that you choose small learning rates (as you write). I am not a big fan of the tactic to rely on the hope that the used, poor, optimizer will not find an obvious flaw in an objective function. Experimental details: - I assume the correct thing to do during evaluation would be to sample from the posterior defined by Equation (3), but this is never mentioned. If this is not what you do, what do you do ? Do you simply use the maximum action from the deterministic network output ? - While the task you evaluate on is interesting and fits the presented model well it would have been nice to see a comparison also on one more standard task in which one would expect hierarchical behavior to help, to facilitate comparison to existing work. This is less of a concern if the used data will be made available with the publication of the method. Regarding the visualizations: - While it indeed seems that your model does not reproduce exact test trajectories it would have been interesting to also depict the closest trajectory in the training data (perhaps aligned with time-warping) to the one produced for the depicted test cases (hinting that the model does not merely learn to remember the training data). - You also could have included more visualizations for the baseline methods as a comparison (at least in the supplementary material). Regarding the Human preference study: - You write that "During extrapolation, the other nine players do not animate.". This seems odd to me. First of all, I assume that you mean that the players are not animated in the visualization while they are still simulated for the network input (as the network should only be capable of dealing with such examples). Second, why is this a sensible idea ? Would this not heavily restrict the capabilities of the experts to tell apart a real, played out, strategy from a generated one ? I somehow doubt that a play can be extrapolated reliably by the experts just based on a few initial frames. I would rather have expected that you simply highlight the target player. Regarding the evaluation of the macro-planner: - From the visualizations in Figure 6b it seems that the agent only rarely visits the actual macro goals. This is surprising to me. Could this be a result of the fine-tuning procedure (in which no macro labels are provided) and hint at the fact that the model indeed does not nicely separate macro and micro actions (as I would suspect it will not given the objective function used, see discussion above) ? Regarding the Benchmark analysis: - The number of different model combinations studied here is nice and it allows for a concrete attribution of performance improvements to different modeling choices. - The evaluation presented in Table 2 has one main problem: You evaluate classification accuracy of the different studied models. However, given that you discretize the two-dimensional continuous action space into a large 17x17 categorical distribution I am not surprised by the generally low accuracies. While I understand that this accuracy is what you optimize during training it is definitely conceivable that a trained model might predict slightly different actions in each step which have almost the same effect as the ground truth actions but result in 0 % classification accuracy. As such, a different (perhaps distance based) measure would have been good to have in this table! - Tying into the question regarding deterministic/stochastic action choices above, it would have also been nice to have an evaluation of stochastic action selection here (which I assume is not what is presented in the table, although this is also not mentioned). - It also would have been better if you had splitted the test-data into several folds and reported mean and variances (or perhaps better medians with quantiles) in the table such that it would be possible to judge the statistical significance of the differences. Minor issues: - The figures in the table are of terrible quality, making the model graphs hard to read. Please convert such graphics to vector graphics or at least include high resolution versions. I don't quite understand how you did not realize this before submission, did something go wrong in the submission process here ? - In-text citations are consistently in the wrong citation style. Please read the author guidelines and correctly use natbib \citet/\citep citations! - There are a few articles missing throughout the text and some other minor text mistakes. Here are the instances I found: l. 22 "timescale low-level actions" -> "timescale of" l. 33 "For instance, in Figure 1" here you are talking about the green "player: but that is not mentioned in the text l. 68 "previous work which focus on" -> "which focuses on" l. 96 "policy should simultaneously optimizes behavior" -> "optimize" l. 176 "we used an simultaneously predicted the next 4" -> "and" l. 184 "but instead learns the final output from a feature" -> "which instead" l. 255 "may applicable" -> "may be applicable"

Confidence in this Review

3-Expert (read the paper in detail, know the area, quite certain of my opinion)